# Histopathological domain adaptation with generative adversarial networks: Bridging the domain gap between thyroid cancer histopathology datasets

William Dee[1]*, Rana Alaaeldin Ibrahim[2,3], Eirini Marouli[1,4]*

1 Digital Environment Research Institute (DERI), Queen Mary University of London, London, United Kingdom, 2 Centre for Oral Immunobiology and Regenerative Medicine, Institute of Dentistry, Barts and The London School of Medicine and Dentistry, Queen Mary University of London, London, United Kingdom, 3 Department of Oral Pathology, Faculty of Dentistry, Mansoura University, Egypt and Queen Mary University of London, London, United Kingdom, 4 William Harvey Research Institute, Barts and The London School of Medicine and Dentistry, Queen Mary University of London, London, United Kingdom

* w.t.dee@qmul.ac.uk (WD); e.marouli@qmul.ac.uk (EM)

**Data Availability Statement:** Source codes, as well as underlying data, are freely available at: https://github.com/williamdee1/ThyCa-GAN.

## Abstract

Deep learning techniques are increasingly being used to classify medical imaging data with high accuracy. Despite this, due to often limited training data, these models can lack sufficient generalizability to predict unseen test data, produced in different domains, with comparable performance. This study focuses on thyroid histopathology image classification and investigates whether a Generative Adversarial Network [GAN], trained with just 156 patient samples, can produce high quality synthetic images to sufficiently augment training data and improve overall model generalizability. Utilizing a StyleGAN2 approach, the generative network produced images with an Fréchet Inception Distance (FID) score of 5.05, matching state-of-the-art GAN results in non-medical domains with comparable dataset sizes. Augmenting the training data with these GAN-generated images increased model generalizability when tested on external data sourced from three separate domains, improving overall precision and AUC by 7.45% and 7.20% respectively compared with a baseline model. Most importantly, this performance improvement was observed on minority class images, tumour subtypes which are known to suffer from high levels of inter-observer variability when classified by trained pathologists.

## Introduction

Thyroid cancer incidence has generally been increasing since the 1970s [1]. It is currently the ninth most common cancer worldwide [2], and the 2019 Global Burden of Disease study predicted incidence will increase across all age groups for the next 20 years [3].

Differentiated thyroid cancer (DTC) includes all types of thyroid cancer that originate in the cells which produce and store thyroid hormones, and accounts for approximately 90% of thyroid cancer incidence [3, 4]. DTC generally has a good prognosis compared with

**Funding:** The author(s) received no specific funding for this work.

**Competing interests:** The authors have declared that no competing interests exist.

undifferentiated thyroid cancers which include anaplastic and medullary thyroid cancer [5]. Within this DTC designation, the most common thyroid gland malignancy is papillary thyroid carcinoma (PTC), constituting 80–90% of diagnosed cases [3, 6]. Notably, several genetic mutations have been implicated in the pathogenesis of PTC especially BRAF V600E [7]. The ALK gene mutation has been observed in various thyroid cancers, including PTC, follicular thyroid carcinoma (FTC), and undifferentiated anaplastic thyroid cancer [8]. Additionally, the C228T promoter mutation in the TERT gene is associated with the BRAF V600E mutation in PTC [9, 10].

Other genetic mutations reported in various thyroid cancer include RAS [11, 12] PIK3CA, AKT1, PTEN [13], mTOR [14], and chromosomal rearrangements involving RET/PTC and PAX8-PPARγ genes [12]. These genetic mutations can deregulate the mitogen-activated protein kinase (MAPK) and phosphatidylinositol-3 kinase (PI3K)/AKT signalling pathways which are crucial to the pathogenesis of thyroid cancer [11]. It is noteworthy to mention that MAPK activation is crucial for the initiation of PTC [13]. Other signalling pathways include p53 and Wnt/β-catenin [13].

Distinctive nuclear features aid in the clinical classification of PTC and its variants designated as PTC-like [15–20]. These include changes to nuclear size and shape, primarily elongation, enlargement and overlapping nuclei, as well as chromatin alterations such as "clearing, margination and glassy nuclei" [21, 22].

Whilst histopathology assessments remain the gold standard in tumour diagnosis [15, 23], there still exists significant inter-observer variability between diagnoses [24, 25]. Additionally, diagnostic accuracy is dependent on the experience of the pathologist [26], and greater patient imaging throughput is placing increasing demands on the time of these highly qualified professionals [27].

In the past two decades, the evolution of whole-slide image technology has facilitated the digital storage of high-resolution histopathological images. The ability to share high quality sample data globally has enabled the development of various machine learning approaches aimed at automating histopathological image classification. Computer-based methods have the potential to improve diagnosis speed and accuracy, as they require less overall training time than a human and have been shown to outperform experienced clinicians in various image classification tasks [28, 29]. Furthermore, an interpretable ML system can be used to aid pathologist training, as well as enabling quality assurance and the assessment of both inter and intra-observer variability [23].

Accurate diagnosis is particularly important within thyroid cancer as overdiagnosis is a known and growing issue, accounting for up to 60–90% of newly diagnosed cases [30, 31]. This overdiagnosis places needless psychological burden on patients and can lead to overtreatment, i.e., unnecessary thyroidectomy surgery [32].

Due to PTC's prevalence and relatively clear features, combined with long-term survival rates of more than 90% [33], identifying patient samples with PTC-like nuclei is the first step in a pathologist's diagnostic approach [34]. Machine learning approaches have therefore often focused on automating the bulk of the diagnosis burden, classifying histopathological images as PTC-like or not [18, 27, 34–36]. These methods have utilised a variety of architectures, ranging from Random Forests and Support Vector Machines (SVMs) [37–39] to deep learning Convolutional Neural Network (CNN) models [34, 40, 41].

Böhland et al. [34] directly compared seven different machine learning-based approaches for predicting the presence of PTC-like nuclei in whole-slide image patches. While the best performing method achieved 89.7% accuracy when tested on set-aside data from the same domain as the training data, it classified minority class non-PTC-like samples, sourced from a separate domain, with 46.7% accuracy.

The authors suggested this failure to generalize was due to a lack of diverse training data, a common problem when applying machine learning methods to (often small) medical imaging datasets [28]. This issue can be exacerbated when using deep learning, which typically requires large datasets to produce highly generalizable models [42, 43]. A lack of training data diversity can be especially problematic when there is a large domain gap present between datasets, i.e., differences in data distributions and/or feature representations caused by the underlying processes behind gathering and processing the data. These domain differences can obscure the underlying biological information, making it difficult to train robust models which can adapt well to the new data.

A Uniform Manifold Approximation and Projection (UMAP) [44] plot can be used to visualize the domain gap between datasets. Images from the Tharun and Thompson (T&T) [34] and Nikiforov datasets [45] used in Böhland et al.'s work were passed through a pretrained ResNet50 [46] model to obtain embeddings, before being represented in a two-dimensional latent space using UMAP (Fig 1). The separation between the two sets of embeddings demonstrates that the data distributions are clearly distinct from each other—i.e., there are stronger differences caused by the domain origination, than similarities relating to samples which share the same histopathological classifications.

## There are no sources in the current document

One potential solution to bridge this domain gap is to apply generative adversarial networks (GANs). GANs were introduced by Goodfellow et al. [47] as a method of producing high quality synthetic (fake) images which approximate an underlying real data distribution. Adversarial training methods have previously been applied successfully to generate artificial data in MRI reconstruction and tumour segmentation [48], X-ray organ segmentation [49, 50], and virtual slide staining [51–53].

GAN-generated synthetic images have proven to be convincing representations when presented to trained medical professionals. Synthetic lung nodule samples were assessed as real 67% of the time by a radiologist with 13 years' experience, whilst 100% were considered real by a radiologist with four years' experience [54]. Both board-certified and trainee pathologists showed an inability to distinguish between real and synthetic ovarian carcinoma samples, choosing correctly only 54% of the time [55]. Lastly, Xue et al. [52] found that three out of four pathologists could not differentiate over half of the synthetic histopathology images produced by their GAN. Furthermore, augmenting training data with GAN-generated images has been shown to increase the performance of machine learning models when generalizing to unseen test data [56, 57].

In this paper we investigated whether a GAN could be successfully trained on a limited dataset of 156 patient histopathology slides to produce realistic synthetic images. We evaluated the impact of using these GAN-generated images to augment the original training data and measured the improvement when classifying whether held-out test samples from the same domain had PTC-like nuclei present.

In addition, we curated a new dataset, combining publicly available histopathology data from three separate domains (see Methods: Data acquisition and processing for more detail). This dataset contains histological subtypes which are frequently misclassified by trained histopathologists due to their relative scarcity or the complexity of diagnosis criteria. We assessed whether our generated synthetic images helped bridge the domain gap present between the training and test data. We aim to show that improved classification performance can be gained for these difficult minority class subtypes, with generative data augmentation allowing a deep

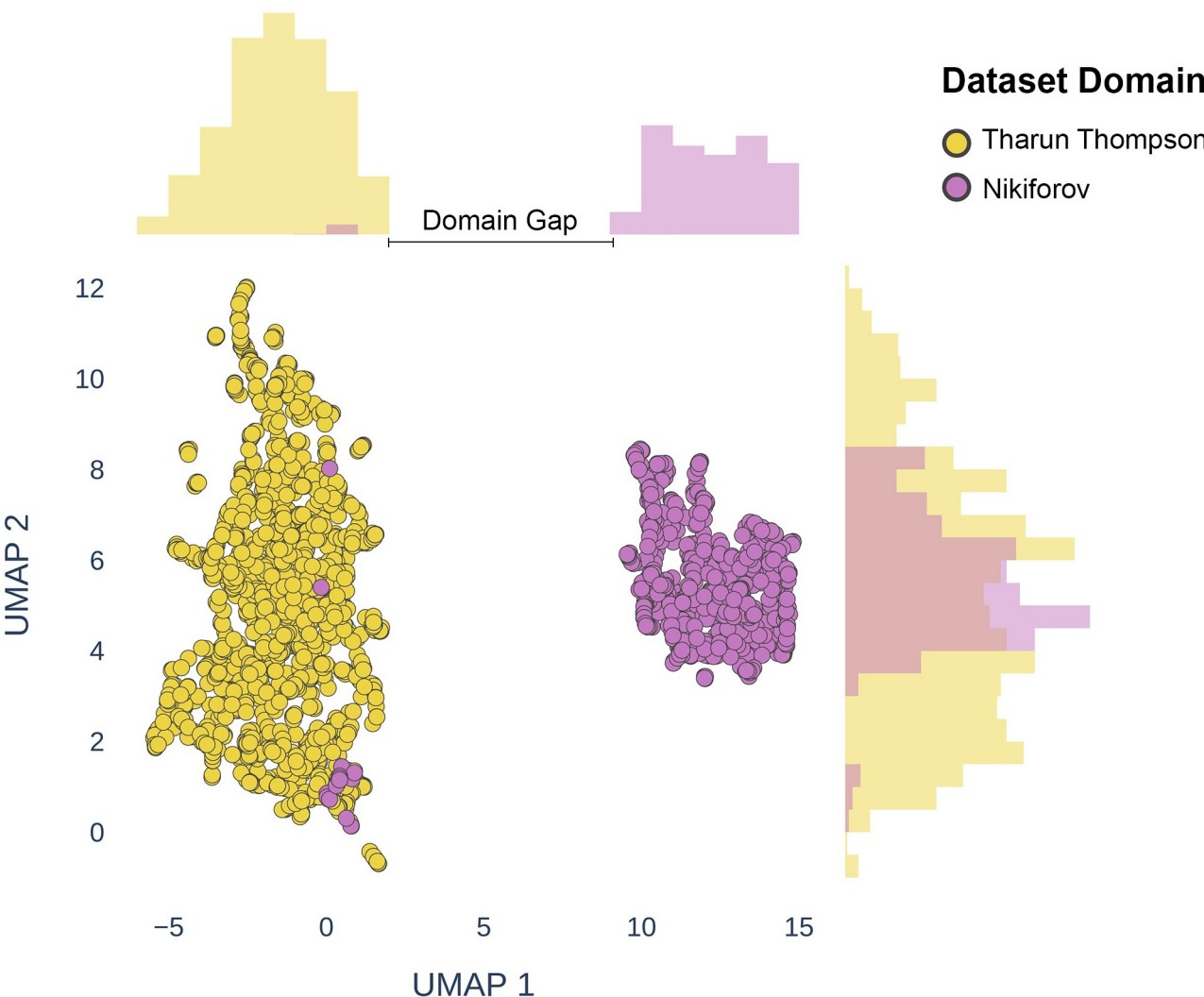

**Fig 1. UMAP visualization of the domain gap between the T&T [34] and the Nikiforov datasets [45].** Images from both datasets were passed through a pretrained ResNet50 [46] model to obtain embedding representations. The figure shows a clear separation between the two datasets along the first UMAP dimension. This gap can have a negative impact on the ability of a model trained on one dataset being able to generalize to predict the other.

learning model to partially overcome domain differences and focus on the biological signal present in histopathological images.

An overview of our approach is included in Fig 2.

## Results

### GAN training

Table 1 shows the results of the StyleGAN2 training. The FID score significantly improves to 5.05 when the original 1,916 x 1,053 px images were split into non-overlapping 512 x 512 px crops to increase the amount of available training data. This score is similar to the 4.67 average score that Karras et al. (2020) [58] obtained for three 5,000 image "Animal Faces (AFHQ)" datasets. It is also notably lower than the FID score of 15.71 Karras et al. (2020) [58] achieved on the BreCaHAD dataset, which consisted of 162 breast cancer histopathology images.

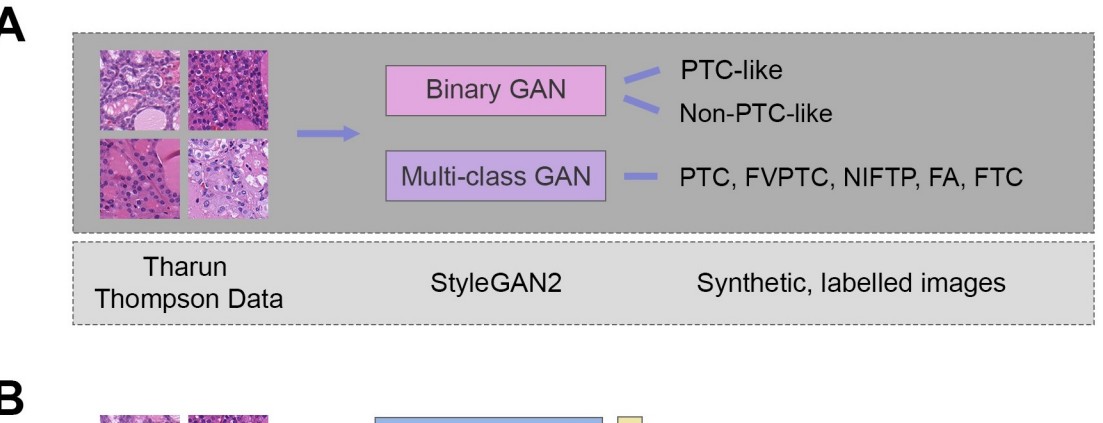

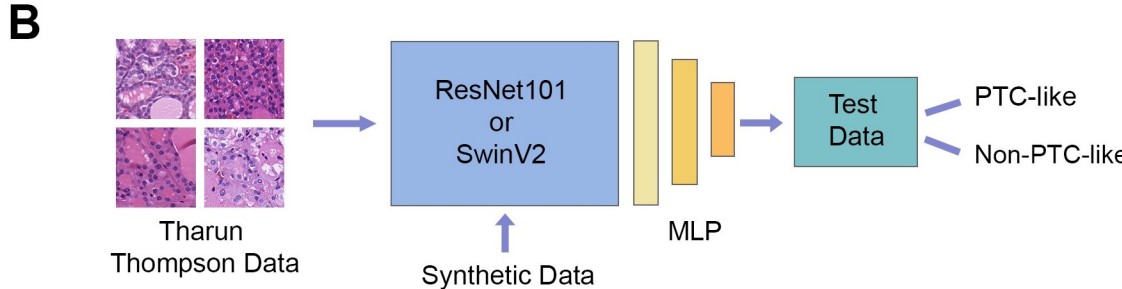

**Fig 2. An overview schematic of the process flow followed by the methods of this paper.** (A) An overview of how either a binary-class or multi-class GAN is trained to produced synthetic samples for a given label using the Tharun Thompson data. (B) Shows how the original training data is combined with the synthetic data to produce a deep learning model which can generalize to unseen test data. Note, synthetic images are only added to the training data in the case that GAN augmentation is being utilized, otherwise (i.e., during baseline model training) no synthetic samples are added to the training data.

Fig 3 depicts synthetic images generated by GANs trained on binary labels. It compares images produced using two different training sets: one consisting of the 1,496 centrally cropped T&T images, producing an FID score of 18.38, and another consisting of 12,038 overlapping crops from the same T&T dataset, resulting in an FID score of 5.10.

## Deep learning classification

**T&T dataset.** Table 2 displays the five-fold cross-validation performance of the model trained to classify images from the T&T dataset. A ResNet101 architecture [59] with a three-

**Table 1. GAN training results.**

| Training Images | Data Labelling | Min. FID |
|:---:|:---:|:---:|
| 1,496 | None | 15.39 |
| 1,496 | Binary | 18.38 |
| 12,038 | Binary | 5.10 |
| 12,038 | Multi-class | 5.05 |

Results from GAN training showing the minimum FID score under different approaches. Where data labels are "None" the GAN has been trained to produce a generic thyroid histopathology image. "Binary" indicates the GAN has been trained conditionally on samples either labelled as PTC-like or non-PTC-like. "Multi-class" denotes that the GAN was trained with conditional labels for each classification subtype, being PTC, NIFTP, FVPTC, FA and FTC.

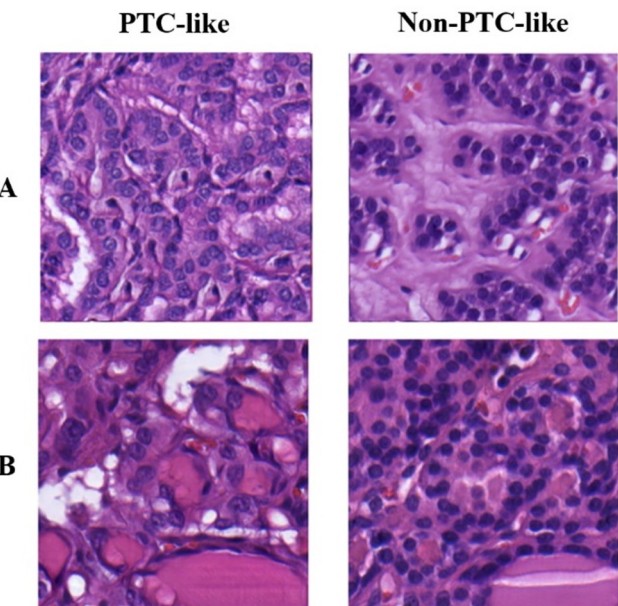

**PTC-like**   **Non-PTC-like**

A

B

**Fig 3.** Examples of PTC-like and non-PTC-like images produced after Style-GAN2 was trained conditionally using: (A) the original 1,496 T&T dataset with 512 x 512 px crops extracted from the centre of each image, achieving an FID score of 18.38. (B) Compared with the expanded T&T dataset which extracted 12,038 overlapping 512 x 512 px crops from the original images, resulting in an FID score of 5.10.

layer multi-layer perceptron was used as the classifier (see Methods: Deep learning classifier (DLC) for more detail).

Our baseline model achieved similar overall accuracy to both approaches in Böhland et al.'s work, classifying datapoints with 1% lower overall accuracy. Notably, the worst predicted class remains NIFTP, one of the minority classes in the data, and a subtype considered different for trained pathologists to identify with high accuracy.

Increasing the number of samples of each binary classification (PTC-like or not) using the binary GAN resulted in an increase in accuracy of 10%, improving across all classes. Using a

**Table 2. Five-fold cross-validated T&T results.**

| Classificati on | PTC-like | | | Non-PTC-like | | Accuracy |
|---|---|---|---|---|---|---|
| | PTC | FVPTC | NIFTP | FA | FTC | |
| *No. Samples* | *53* | *9* | *9* | *53* | *32* | *156* |
| ***Böhland et al.* [34] *Models:*** | | | | | | |
| **FBC** | 90.57 | 88.89 | 66.67 | 98.11 | 81.25 | 89.74 |
| **DLC** | **94.34** | 88.89 | 55.56 | 92.45 | 84.38 | 89.10 |
| **Models produced by this paper:** | | | | | | |
| **DLC** | 80.00 | 90.00 | 70.00 | 100 | 86.66 | 88.13 |
| **Binary GAN** | 98.18 | 100 | 96.67 | 90.00 | **100** | 98.13 |
| **Multi-class GAN** | **98.18** | **100** | **100** | **100** | **100** | **99.38** |

Five-fold cross-validation fold accuracy metrics for DLC models trained to predict samples as PTC-like or non-PTC-like from the T&T dataset. The training of each fold is performed with a 60% split of the data used for training, 20% for validation and 20% for testing. Baseline includes no GAN-generated images within the training data, whilst Binary 100 and MC 100 include synthetic data augmentation as set out in Synthetic data augmentation.

**Table 3. NTE results.**

| Classification | Accuracy | | | | Precision | Recall | AUC |
|---|---|---|---|---|---|---|---|
| | PTC-like | | Non-PTC-like | | | | |
| Subtype | FVPTC | NIFTP | FA | B | Precision | Recall | AUC |
| *No. Samples* | 30 | 25 | *12* | *11* | | | |
| **Baseline (ResNet101)** | 33.33 | 16.00 | **100** | **90.91** | **93.33** | 25.45 | 68.96 |
| **Baseline (SwinV2)** | 50.00 | **92.00** | 91.67 | 9.10 | 77.55 | **69.09** | 64.31 |
| **Binary GAN** | 40.00 | 76.00 | **100** | **63.64** | 85.00 | 61.82 | **71.51** |
| **Multi-class GAN** | **60.00** | 56.00 | 100 | 45.45 | 84.21 | 58.18 | 70.23 |

DLC results when tested on the NTE dataset. The "Baseline" models utilize 80% of the T&T dataset for training and 20% for validation. The "Binary 100" and "MC 100" models both use GAN-generated synthetic samples during training (described in Methods: Synthetic data augmentation), to augment the original T&T training data. The entire NTE data is used as the test set.

multi-class GAN to specifically augment each class, equalizing the number of training examples across classes (see Methods: Synthetic data augmentation for more detail), resulted in the best performance with an accuracy of 99.38%.

**NTE dataset.** Like the five-fold cross-validation results seen with the T&T data, augmenting the training data with GAN-generated samples has a positive impact on performance when tested on the external NTE data (Table 3). The ResNet101 baseline model was adept at classifying negative (non-PTC-like) images but classified the two different subtypes with PTC-like nuclei, FVPTC and NIFTP, with only 33.33% and 16% accuracy respectively. This approach therefore had a low recall score of 25.45%, illustrating the model had not adapted well to the external domain samples.

Using a SwinV2 [60, 61] model in place of ResNet101 resulted in improved recall, to the detriment of precision and recall as benign samples were classified with 9.10% accuracy and overall AUC dropped to 64.31%.

Using the binary and multi-class GAN for data augmentation resulted in more balanced accuracy scores across subtypes, reducing the range of classification performance. This resulted in higher precision scores of 85.00% and 84.21% compared with the SwinV2 baseline of 77.55%. Additionally, both GAN augmentation methods improved the model's AUC over the baseline models, showing a stronger ability for those models to discriminate between PTC-like and non-PTC-like images overall.

## Discussion

In this work we successfully trained a GAN which could produce high quality synthetic thyroid histopathology images using a limited dataset of images from 156 patients. The GAN FID scores were comparable to prior state-of-the-art results for similar-sized datasets from different domains [58], showing the applicability of the StyleGAN2 approach to the medical imaging domain. We demonstrated that including these synthetic samples in the training data for a deep learning model resulted in a more robust model which could generalize more effectively to unseen external data.

The NTE dataset mirrored the key challenges present when considering deploying in-silico models in practice, focusing on the real-world issues of both data scarcity and heterogeneity. Our results support the use of GANs as a method for data augmentation in this field, offering evidence that generative models can learn some of the key characteristics of histopathological

image data by approximating the underlying population distribution, improving classification performance in the absence of more real training images.

Two GAN-augmentation strategies were trialled on two held-out test sets—T&T (via five-fold cross-validation) and the NTE data, which was sourced from multiple domains. In both instances, augmenting the training data with GAN-generated synthetic samples proved to be beneficial.

In the case of the T&T data, classification accuracy increased by 10% with the binary GAN and 11% with the multi-class GAN compared with the baseline created by this paper and the original models from Böhland et al.'s research [34]. Even with only 156 different patient samples the baseline classification accuracy of 88.13% is high, so the additional diversity and number of images added with the GAN augmentation meant the classification was able to achieve a high accuracy across all classes.

Both GAN strategies notably increased the ability of the model to classify the minority PTC-like samples, FVPTC and NIFTP, which were relatively poorly classified by the baseline. These subtypes are much more difficult to identify because the PTC-like features can be found within encapsulated follicular lesions, are less numerous and distinct, and are often surrounded by benign-appearing cells [20, 62–66]. Increasing the frequency of training data via GAN data augmentation appears to enable the model to pick up on these more subtle aspects in the underlying data, resulting in more robust models across all classes.

Despite the high accuracy for the models tested on the T&T data, when the same ResNet101-based model (see Baseline: ResNet101 in Table 3) was trained to classify the NTE dataset it performed poorly at predicting which samples had PTC-like nuclei present. This baseline model classified FVPTC and NIFTP samples with 33.33% and 16% accuracy respectively. Whilst it predicted the non-PTC-like samples with high accuracy, this reflects the model's inability to distinguish between classes, simply classifying most datapoints as non-PTC-like, resulting in a low recall score of 25.45%.

To improve classification performance, the ResNet architecture was switched for a SwinV2 model architecture, which has proven effective within the medical domain for capturing phenotypic differences captured in microscopy images [67, 68]. The baseline SwinV2 model achieved a higher recall score compared with the ResNet approach, however, was unable to differentiate between the NIFTP and benign (B) samples from the Nikiforov data, classifying all but three samples as PTC-like. The model was therefore still unable to bridge the domain gap observed in Fig 1 between the T&T training data and the Nikiforov external domain data included within the NTE test dataset.

Augmenting the training data using either the GAN trained to produce binary class samples or multi-class samples improved the stability of classification accuracy across subtypes. The binary GAN showed the greatest ability to differentiate between the PTC-like (NIFTP) and non-PTC-like (B) samples from the Nikiforov data, classifying them with 76% and 63.64% accuracy respectively.

Both models trained with GAN-augmented data achieved higher precision scores compared with the SwinV2 baseline. Precision is particularly important within thyroid diagnosis given the issues in the field with overdiagnosis and the patient and treatment burden associated with false positive results [30–32]. Additionally, the AUC scores for both GAN-augmented models improved by 7.20% (binary) and 5.92% (multi-class) over the SwinV2 baseline, demonstrating their increased classification robustness across subtypes within the binary class designations.

The NTE dataset was created as a difficult test for a model to be able to generalize, given the range of demographics, batch effects and other domain differences across its samples. Interestingly, the FA class is well-predicted across all models, however Fig 4 (see Methods: NTE

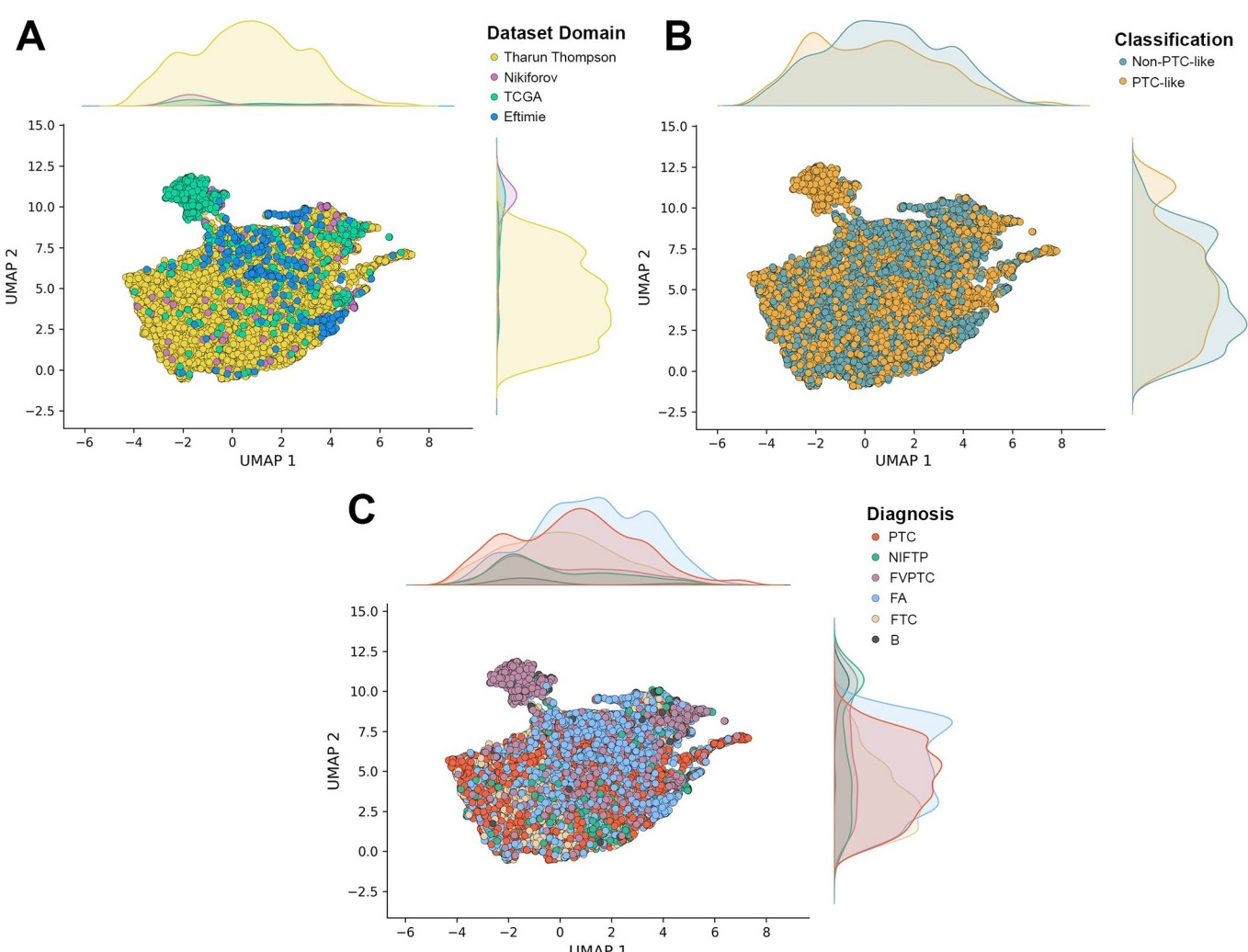

**Fig 4. Images from the T&T and NTE dataset were passed through a pre-trained ResNet101 classifier to produce embeddings.** These embeddings were then visualized using the Uniform Manifold Approximation (UMAP) and coloured according to: (A) dataset domain, referring to the paper which produced and shared the original images; (B) classification as having PTC-like or non-PTC-like nuclei present; (C) subtype diagnosis. As can be seen in (A) the Eftimie FA samples are more similar to the T&T data, as represented by their closeness in two-dimensional space, than the Nikiforov or TCGA samples are, models with therefore likely generalize better from the T&T data to the Eftimie data than they will to the Nikiforov or TCGA data.

dataset) shows the Eftimie data (FA samples) is more similar to the T&T training data than the Nikiforov (NIFTP, B) or TCGA (FVPTC) data. Thus, the model is better able to generalize to this data.

We have shown in our work that without generative data augmentation our models fail to generalize well to external data where there are significant domain differences—i.e., the Nikiforov and TCGA samples within the NTE dataset. Whilst classification accuracy is not high across all subtypes these are different classifications for even a trained pathologist to make, so any increase in a model's precision or AUC represents an important contribution to the field.

For future work, an alternative solution to bridging the domain gap between datasets could be to use a style-transfer GAN. This approach aims to separate images into "content" and "style" spaces [69]. In the context of this report, content would represent the various structural features of the histopathology images, whilst the domain differences, such as the stain colour

or resolution, would be considered the style. Additionally, creating class-specific connected generative models could increase accuracy across the minority subtypes.

Finally, one criticism of deep learning architectures is that they function like "black box" models. Explainable AI is particularly important in healthcare as accountability, trust and understanding bias are integral components to the functioning of the system as a whole [59, 70]. Using a self-attention GAN (SAGAN) [71] could partially alleviate these issues. The attention mechanism [72] can be visualized to provide feedback regarding which parts of the image were most important for the model's classification decision. This could form part of an important feedback loop for a pathologist to understand and interpret the model output, increasing trust in its ability to accurately classify pathological or other medical images.

# Materials and methods

## Data acquisition and processing

**Tharun and Thompson ("T&T") dataset.** The dataset is comprised of 156 patient whole slide images (WSIs) of thyroid gland tumours, 138 sourced from the University Clinic Schleswig-Holstein, Campus Luebeck, and 18 from the Woodland Hills Medical Centre, California [34]. Two pathologists agreed on the classification of each tumour, before 1,916 px by 1,053 px crops were extracted from the identified neoplastic regions of interest. Each image has an objective magnification factor of 40x and a resolution of 0.23 μm/px. The dataset was requested by emailing sekretariat.patho@uksh.de. See Table 4 for additional information.

PTC samples constitute the majority of the positive "PTC-like" class, whilst the FVPTC and NIFTP subtypes, which are considered much more difficult to diagnose due to their less distinctive nuclear features [34], are minority samples within the data—mimicking their real-life comparative scarcity. The "Non-PTC-like" class consists of the two most common diagnoses which lack PTC-like nuclei—FA and FTC. Additional detail about these classifications can be found in the S1 File.

**NTE dataset.** The NTE dataset (Table 5) is comprised of WSIs from three separate domains. Firstly, 36 samples (25 non-invasive follicular thyroid neoplasm with papillary-like nuclear features, 11 benign) were obtained from "Box A" of the Nikiforov online repository, relating to research performed by Nikiforov et al. [45]. The study accepted WSI contributions from 13 institutions across six different countries, before a panel of 24 expert thyroid pathologists determined each slide's classification. The research sought to establish consensus diagnostic criteria for classifying NIFTP as a separate subcategory and therefore accepted many borderline cases which were considered difficult to diagnose even by expert pathologists [34, 45]. The images were processed at a resolution of 0.49 μm/ px and magnification of 40x.

**Table 4. Summary of the T&T dataset.**

| Subtype | Classification | Source Data | Patients |
|---|---|---|---|
| Papillary thyroid carcinoma (PTC) | PTC-like | T&T | 53 |
| Follicular variant of papillary thyroid carcinoma (FVPTC) | PTC-like | T&T | 9 |
| Non-invasive follicular thyroid neoplasm with papillary-like nuclear features (NIFTP) | PTC-like | T&T | 9 |
| Follicular thyroid adenoma (FA) | Non-PTC-like | T&T | 53 |
| Follicular thyroid carcinoma (FTC) | Non-PTC-like | T&T | 32 |
| **Total** | | | **156** |

Sample subtypes present in the Tharun and Thompson ["T&T"] dataset, including their classification as PTC-like or not and the number of patient samples in each category.

**Table 5. Summary of the NTE dataset.**

| Subtype | Class | Data Source | Patients |
|---|---|---|---|
| Follicular variant of papillary thyroid carcinoma (FVPTC) | PTC-like | TCGA [73] | 30 |
| Non-invasive follicular thyroid neoplasm with papillary-like nuclear features (NIFTP) | PTC-like | Nikiforov [45] | 25 |
| Benign (B) | Non-PTC-like | Nikiforov [45] | 11 |
| Follicular thyroid adenoma (FA) | Non-PTC-like | Eftimie [15] | 12 |
| **Total** | | | **78** |

31 follicular variant of papillary thyroid carcinoma (FVPTC) samples were sourced from The Cancer Genome Atlas (TCGA) Thyroid Carcinoma study. This study consists of 507 different patient samples, collected from 20 separate tissue source sites. Each patient was originally diagnosed with PTC, before a board-certified pathologist assigned fine-grained subtyping to each sample [73]. Within the TCGA data there are a range of magnifications and resolutions, according to the equipment used at the source site. Finally, 12 follicular thyroid adenoma (FA) samples were obtained by emailing the corresponding author of Eftimie et al. [15]. These were originally imaged using a 20x magnification (see S2 File for further information regarding sample selection).

The NTE dataset was formed specifically to provide a robust test of a model's generalizability. The FVPTC and NIFTP subtypes are commonly misdiagnosed by expert clinicians, and therefore are most important to be able to predict. These two subtypes account for 9 samples each within the T&T training dataset (Table 4), mimicking the severe lack of training data which is often prohibitive to highly generalizable models in practice. Lastly, the NTE dataset has high heterogeneity in terms of staining, resolution, magnification, and image quality, due to the combination of WSI patches from multiple different domains each with differing instruments and collection procedures. Across the data sources there are also a range of patients from diverse hospitals with varied demographics.

Sample subtypes present in the NTE dataset, comprised of images sourced from the research by Nikiforov et al. [45], Eftimie et al. [15] and the TCGA Thyroid Carcinoma study [73]. The table includes each sample's binary classification as having PTC-like nuclei or not as well as the number of patient samples for that designation.

Fig 5 shows PTC-like and non-PTC-like examples from both the T&T and Niki-TCGA datasets, displaying the range of staining and resolution present.

**NTE data pre-processing.** WSIs for each sample in the dataset were downloaded, before a trained pathologist identified the neoplastic regions of interest within each slide which were indicative of the underlying classification (S3 File). From these regions, the OpenSlide function 'DeepZoomGenerator' was used to extract non-overlapping 512 x 512 px crops, before 20 from each sample were selected at random for inclusion within the dataset.

## Generative Adversarial Network (GAN)

GANs are comprised of two neural network models, referred to as the generator and the discriminator. The generator aims to learn to approximate the underlying distribution of the training data to generate high-quality synthetic images. The discriminator conversely learns to discern the difference between these GAN-generated fakes and the real images [47].

The two networks compete during training to improve at their respective roles, producing realistic synthetic examples and detecting fake images. An equilibrium is reached when the

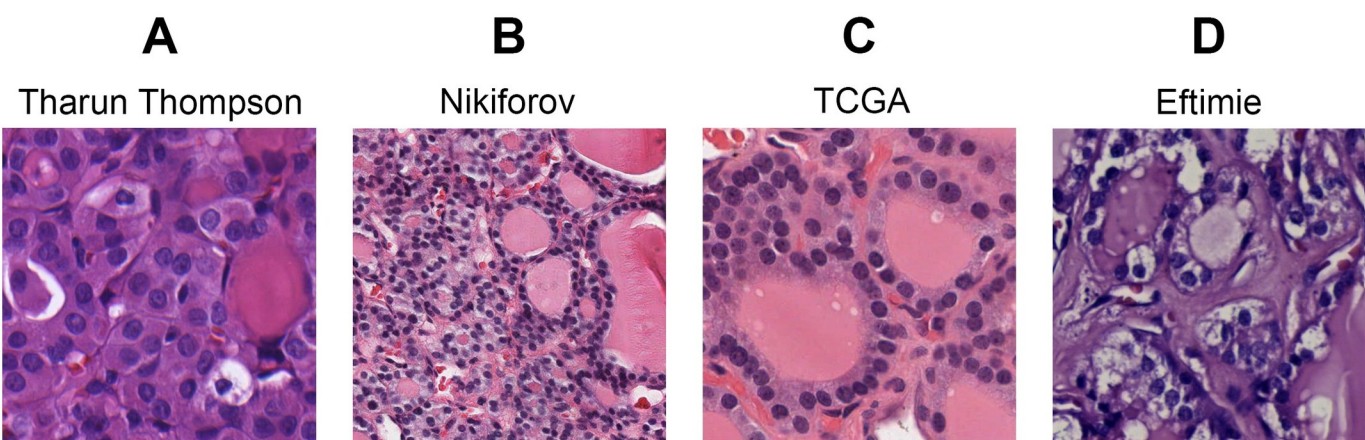

**Fig 5. Example 512 x 512 px image crops taken from the four different sources which comprise the T&T and the NTE datasets.** The classification and dataset identifier of each sample is as follows: (A) T&T—Papillary thyroid carcinoma (Dataset ID: 47h_5), (B) Nikiforov—Noninvasive follicular thyroid neoplasm with papillary-like nuclear features (Dataset ID: NIK-A079_0), (C) TCGA—Papillary carcinoma, follicular variant (Dataset ID: TCGA-EM-A4FH_17), (D) Eftimie— Follicular thyroid adenoma (Dataset ID: EFT-552053_6).

generator is producing images which are indistinguishable from the underlying real data, and thus the discriminator can make predictions about whether an image is real or fake with only 50% accuracy.

The StyleGAN2 framework [74] was selected because it was specifically designed to produce high quality synthetic images from limited training data. It utilizes an extensive array of 18 transformations to augment the discriminator network's training data. The method then adapts the level of augmentation based on feedback from an overfitting heuristic during model training.

The PyTorch implementation of StyleGAN2 was adapted from the official StyleGAN2-ADA GitHub repository and trained with the following parameters:

- "cfg = paper512" to mirror the parameter settings used by Karras et al. (2020) [58] for the BRECAHAD dataset—a small dataset containing 162 breast cancer histopathology images [75].

- "cond = 1" ensures the GAN is trained conditionally using the labels provided, and so is subsequently able to produce images for a given class.

- "mirror = 1" includes x-flips of each image in the dataset, effectively doubling the training images.

- "kimg = 25000" trains the GAN for up to 25 million training steps. All GANs tested in the StyleGAN2 paper were shown to produce their highest quality images before this point in training.

The GAN was trained with two different conditional labelling approaches to assess the quality and diversity of the synthetic images produced:

- Binary: training samples were labelled to be either PTC-like (1.0 label) or non-PTC-like (0.0).

- Multi-class: labelled according to their individual subtypes, being PTC (0.0 label), NIFTP (1.0), FVPTC (2.0), FA (3.0) and FTC (4.0). This enables the trained GAN to produce synthetic images of any given subtype.

**GAN pre-processing and evaluation.** The original images in the T&T dataset are 1916 x 1053 px. The GAN was trained to produce 512 x 512 px image patches, which are subsequently used to augment training data for a deep learning classifier (DLC) to classify images of the same size. To ensure that the maximum data was made available to train the GAN, the T&T images were split into equal 512 x 512 px patches with minimal overlap. This increased the number of training images from 1,496 to 12,038.

GAN training can be notoriously unstable, as the models are prone to overfitting and mode collapse [55]. To assess progression, batches of synthetic images were manually assessed at fixed intervals and compared to real images. Additionally, Fréchet Inception Distance (FID) [76] was used as the programmatic evaluation criteria for the Generator model. FID computes the Fréchet Distance [77] between multivariate Gaussian approximations of both the real and generated image distributions.

A lower FID score implies a greater alignment between the synthetic and real images and has been correlated with human judgement regarding image quality [78]. Despite this, the metric is not considered perfect, and several papers have warned against over-reliance on FID to assess GAN improvement [58, 78].

In this paper FID was only used as the target metric for GAN training. The true assessment of the quality of the generated synthetic images will be whether their addition to the real images during training improves the classification performance of a deep learning model when tested on unseen data. Our research therefore assesses the impact of GAN samples in an applied scenario, rather than purely evaluating the generative technique in isolation.

## Deep Learning Classification (DLC) model

The DLC model is based on the research performed by Böhland et al (2021) [34] using the T&T dataset. A non-pre-trained version of ResNet101 [79], or SwinV2 [61] was loaded using the Torchvision module, and the final output layer was replaced by a three-layer multi-layer perceptron (MLP) model which outputs a binary prediction—whether the image is PTC-like or not. The model was then trained using the histopathology images in the T&T training set. The Albumentations package was used to apply random cropping, flipping, adaptive histogram equalization, blurring, Gaussian noise, and Fourier Domain Adaptation [80] transformations.

A grid search was used to find optimal parameters for the model. This included an initial learning rate of 1e-3, with a decay of 5e-1 if the validation loss does not decrease for 10 epochs. Early stopping was set at 50 epochs while the model trained for a maximum of 100 epochs. The Adam optimizer [81] and cross-entropy loss were also used. This setup utilized one GPU and a batch size of 32 to obtain the most stable training results.

Five-fold cross validation was used to evaluate the model's performance on the T&T dataset. Each data split contained 60% for training, 20% for validation and 20% for testing. The splits were shuffled and stratified according to classification subtype, ensuring that slide patches from the same patient were retained within the same split to avoid data leakage. There are 156 patients in the T&T data, but each patient has multiple image slides (with the same diagnosis) associated with them. The accuracy metrics are therefore calculated at a patient level, rather than at a slide level. The final classification is determined by majority voting, in the case of a 50:50 split decision between the slides, the wrong class is assigned to the patient.

To evaluate the model's generalizability to the external NTE test dataset, the model is trained using 80% of the original T&T data, with the remaining 20% used as the validation set. Precision, recall and Area under the curve (AUC) are used as an additional metrics due to the

subtype class imbalance between PTC-like and non-PTC-like samples and the designations within the NTE dataset between domains. In both instances, if GAN-generated samples are used, they are included within the training data only.

### Synthetic data augmentation

The binary GAN approach uses a GAN trained with binary classification labels (PTC-like or non-PTC-like) and increased the frequency of each of these binary class images by 100% in the training data, thus retaining the original subtype class imbalance.

The multi-class GAN was trained to produce images of all five subtypes in the T&T training data. This method augments the subtype with the most real images (FA) by 100%, and then equalizes all other subtypes to that level, resulting in all classification subtypes having an equal number of training images.

### Supporting information

**S1 File. Additional information regarding subtype classifications.** Detailed information pertaining to the Thyroid tumour classification subtypes which are relevant to this report, including WHO 2022 designations.
(DOCX)

**S2 File. NTE dataset selection.** Additional details about the selection of the samples included within the NTE dataset.
(DOCX)

**S3 File. Neoplastic region identification.** An overview of the selection of neoplastic regions from the downloaded WSIs.
(DOCX)

**S4 File. GAN augmentations.** Tables displaying the amount of GAN-generated synthetic samples included during model training for each augmentation strategy included within the report.
(DOCX)

### Acknowledgments

The authors would like to thank Ryan Reavette for his guidance and feedback during the project, as well as the initial inspiration for the topic on GANs. We would also like to thank Jamie Holdstock for his contribution towards developing 'ripsvs', a tool to download image patches from online-hosted Aperio ImageScope WSIs.

### Author Contributions

**Conceptualization:** William Dee, Eirini Marouli.

**Data curation:** William Dee, Rana Alaaeldin Ibrahim.

**Formal analysis:** William Dee, Rana Alaaeldin Ibrahim.

**Investigation:** William Dee, Rana Alaaeldin Ibrahim.

**Methodology:** William Dee.

**Resources:** William Dee.

**Software:** William Dee.

**Supervision:** Eirini Marouli.

**Validation:** William Dee.

**Visualization:** William Dee.

**Writing – original draft:** William Dee.

**Writing – review & editing:** William Dee, Rana Alaaeldin Ibrahim, Eirini Marouli.

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
