## [Decision Letter · Decision Letter 0]

7 Feb 2024

PONE-D-23-30850Histopathological domain adaptation with generative adversarial networks: Bridging the domain gap between thyroid cancer histopathology datasetsPLOS ONE

Dear Dr. Dee,

Thank you for submitting your manuscript to PLOS ONE. After careful consideration, we feel that it has merit but does not fully meet PLOS ONE’s publication criteria as it currently stands. Therefore, we invite you to submit a revised version of the manuscript that addresses the points raised during the review process.

We look forward to receiving your revised manuscript.

Kind regards,

Avaniyapuram Kannan Murugan, M.Phil., Ph.D.

Academic Editor

PLOS ONE

Journal Requirements:

4. We notice that your supplementary figures are included in the manuscript file. Please remove them and upload them with the file type 'Supporting Information'. Please ensure that each Supporting Information file has a legend listed in the manuscript after the references list.

Additional Editor Comments:

The study by William et al., is interesting and this was also reflected in reviewers' comments.

However, the reviewers raise some important concerns and those points to be addressed before considering it for publication.

In addition, for the general readers, and young researcher in this thyroid cancer field, it would be interesting to add some molecular background information in the introduction.

Particularly, these seminal articles on BRAF, TERT, RAS, ALK, mTOR, etc., and other can be cited (PMID: 27387551; PMID: 23766237; PMID: 25024077; PMID: 21596819; PMID: 30918329).

Reviewers' comments:

Reviewer's Responses to Questions

**Comments to the Author**

1. Is the manuscript technically sound, and do the data support the conclusions?

Reviewer #1: Yes

Reviewer #2: Yes

Reviewer #3: Yes

Reviewer #4: Yes

2. Has the statistical analysis been performed appropriately and rigorously? 

Reviewer #1: No

Reviewer #2: Yes

Reviewer #3: Yes

Reviewer #4: Yes

3. Have the authors made all data underlying the findings in their manuscript fully available?

Reviewer #1: Yes

Reviewer #2: Yes

Reviewer #3: No

Reviewer #4: Yes

4. Is the manuscript presented in an intelligible fashion and written in standard English?

Reviewer #1: Yes

Reviewer #2: Yes

Reviewer #3: Yes

Reviewer #4: Yes

5. Review Comments to the Author

Reviewer #1: It would be better if you wrote it using the information you discovered.

methodology part and result parts that authors hire an editor to assist with any unclear or grammatical issues that arise throughout the text.

A few standardised grammatical and phrase construction changes are required before publication.

Reviewer #2: good job, the paper presents an important problem and a relevant methodological approach within a line of research that offers at times reason for discussion in terms of explainability more than efficacy or effectiveness.

Beyond the descriptions reported on S1, I remain curious to know whether for subtypes detection further genotype/phenotype data integration can be combined into the approach to validate or robustify the observed improvements. Are the authors considering practical steps in such directions? If so, can they add comments to their discussion part?

Reviewer #3: The manuscript presents a comprehensive study on the use of Generative Adversarial Networks (GANs) to generate synthetic thyroid histopathology images to augment a limited dataset for improving the generalizability of deep learning classifiers (DLCs) in medical imaging. The key focus is on addressing the challenges of data scarcity and heterogeneity in thyroid cancer diagnosis, particularly differentiated thyroid cancer (DTC) and its most common subtype, papillary thyroid carcinoma (PTC), along with more challenging minority subtypes like NIFTP and FVPTC.

1. Improvement of image quality in Figures

• It might be better to move Fig 5 and Tables 4-5 to the supplementary material due to redundancy.

• It would be preferable for Fig 6 to appear earlier, from the method section to the results.

• The quality of the figures needs to be improved.

2. Domain gap bridging

• In Fig 1, consider adding a figure that better illustrates the bridging effect. Currently, the figure seems to only show batch effects, making it less meaningful.

3. Performance on Niki-TCGA dataset

• The final performance on the Niki-TCGA dataset is quite poor, which may indicate limited generalizability. What could be the reason for the low prediction performance on the Niki-TCGA set even for PTC-like cases?. Could there be a difference in performance depending on the number of GAN-generated images?

4. Sharing GAN-generated images

• It could be beneficial to share the GAN-generated images along with their FID scores to provide a clearer understanding of the image quality and relevance.

5. F1 scores for MC model

• For the MC model, it would be useful to present the F1 scores for each class individually to provide a more detailed performance evaluation.

Overall, the manuscript presents a promising approach to leveraging advanced machine learning techniques to improve the accuracy and generalizability of diagnostic models in thyroid cancer histopathology, with potential for impacting clinical practice.

Reviewer #4: Suggesting a mention of the S5 figure in the creation of the Nikki-TCGA dataset to improve understanding. Additionally, ensure that the upper and lower figures in S5 have the same UMAP positions for better clarity.

Recommend providing an explanation of Formula 1 and detail the terms used to enhance reader comprehension

Suggest considering more robust metrics than FID to evaluate the model’s performance.

Provide an explanation of the differences on pictures from the datasets. And lack of explanation on the differences between datasets.

Consider explaining why the 100 value was chosen on the data augmentation, and not any other potential number.

Explore the possibility of connecting GANs to improve performance.

6. PLOS authors have the option to publish the peer review history of their article (what does this mean?). If published, this will include your full peer review and any attached files.

Reviewer #1: **Yes: **Shemsu Oumer Ebro

Reviewer #2: **Yes: **Enrico Capobianco

Reviewer #3: No

Reviewer #4: No

---

## [Author Response · Author response to Decision Letter 0]

8 Aug 2024

Thank you very much for all the feedback and suggested revisions. We agreed that the research required additional work and so we have changed our approach extensively, re-writing the entire code base from the ground up and performing adaptations to alter the model architectures to be more effective than when the research was submitted initially. Hopefully you will agree that we have built more robust models which demonstrate a greater ability to generalize to unseen test data produced in different domains. Additionally, we have now sourced data from an additional domain, with the aim of making the results more wide-reaching and showing one model's ability to generalize across multiple domains, and how generative data augmentation can help to enable this - we believe this research will be well received at a time when generative models are increasingly showing applicability across a variety of different domains.

---

## [Decision Letter · Decision Letter 1]

2 Sep 2024

Histopathological domain adaptation with generative adversarial networks: Bridging the domain gap between thyroid cancer histopathology datasets

PONE-D-23-30850R1

Dear Dr. William,

We’re pleased to inform you that your manuscript has been judged scientifically suitable for publication and will be formally accepted for publication once it meets all outstanding technical requirements.

Kind regards,

Avaniyapuram Kannan Murugan, M.Phil., Ph.D.

Academic Editor

PLOS ONE

Additional Editor Comments (optional):

Reviewers' comments:

Reviewer's Responses to Questions

**Comments to the Author**

1. If the authors have adequately addressed your comments raised in a previous round of review and you feel that this manuscript is now acceptable for publication, you may indicate that here to bypass the “Comments to the Author” section, enter your conflict of interest statement in the “Confidential to Editor” section, and submit your "Accept" recommendation.

Reviewer #2: All comments have been addressed

2. Is the manuscript technically sound, and do the data support the conclusions?

Reviewer #2: Yes

3. Has the statistical analysis been performed appropriately and rigorously? 

Reviewer #2: Yes

4. Have the authors made all data underlying the findings in their manuscript fully available?

Reviewer #2: Yes

5. Is the manuscript presented in an intelligible fashion and written in standard English?

Reviewer #2: Yes

6. Review Comments to the Author

Reviewer #2: (No Response)

7. PLOS authors have the option to publish the peer review history of their article (what does this mean?). If published, this will include your full peer review and any attached files.

Reviewer #2: **Yes: **Enrico Capobianco

---

## [Editor Report · Acceptance letter]

11 Sep 2024

PONE-D-23-30850R1 

PLOS ONE

Dear Dr. Dee, 

I'm pleased to inform you that your manuscript has been deemed suitable for publication in PLOS ONE. Congratulations! Your manuscript is now being handed over to our production team.

Kind regards, 

on behalf of

Dr. Avaniyapuram Kannan Murugan 

Academic Editor

PLOS ONE